

# Soil respiration of a Moso bamboo forest significantly affected by gross ecosystem productivity and leaf area index in an extreme drought event

Yuli Liu[1,2,3], Guomo Zhou[1,2,3], Huaqiang Du[1,2,3], Frank Berninger[2,4], Fangjie Mao[1,2,3], Xuejian Li[1,2,3], Liang Chen[1,2,3], Lu Cui[1,2,3], Yangguang Li[1,2,3] and Di'en Zhu[1,2,3]

[1] State Key Laboratory of Subtropical Silviculture, Zhejiang Agricultural and Forestry University, Lin'an, Hangzhou city, Zhejiang province, China

[2] Key Laboratory of Carbon Cycling in Forest Ecosystems and Carbon Sequestration of Zhejiang Province, Zhejiang Agricultural and Forestry University, Lin'an, Hangzhou city, Zhejiang province, China

[3] School of Environmental and Resources Science, Zhejiang Agricultural and Forestry University, Lin'an, Hangzhou city, Zhejiang province, China

[4] Department of Forest Ecology, University of Helsinki, Helsinki, Finland

Corresponding authors
Guomo Zhou, zhougm@zafu.edu.cn
Huaqiang Du, dhqrs@126.com

## ABSTRACT

Moso bamboo has large potential to alleviate global warming through carbon sequestration. Since soil respiration ($R_s$) is a major source of $CO_2$ emissions, we analyzed the dynamics of soil respiration ($R_s$) and its relation to environmental factors in a Moso bamboo (*Phllostachys heterocycla cv. pubescens*) forest to identify the relative importance of biotic and abiotic drivers of respiration. Annual average $R_s$ was 44.07 t $CO_2$ ha$^{-1}$ a$^{-1}$. $R_s$ correlated significantly with soil temperature ($P < 0.01$), which explained 69.7% of the variation in $R_s$ at a diurnal scale. Soil moisture was correlated significantly with $R_s$ on a daily scale except not during winter, indicating it affected $R_s$. A model including both soil temperature and soil moisture explained 93.6% of seasonal variations in $R_s$. The relationship between $R_s$ and soil temperature during a day showed a clear hysteresis. $R_s$ was significantly and positively ($P < 0.01$) related to gross ecosystem productivity and leaf area index, demonstrating the significance of biotic factors as crucial drivers of $R_s$.

## INTRODUCTION

Soils are important sources and sinks in the global carbon budget (*Sheng et al., 2010*). Soil respiration ($R_s$) is a major source of $CO_2$ emissions from terrestrial ecosystem, and as the second largest carbon flux between the atmosphere and ecosystems it is surpassed only by gross primary production (*Raich & Potter, 1995*). Soils release approximately 75–100 Pg C per year globally (*Bond-Lamberty & Thomson, 2010*), nearly 10 times of the amount of $CO_2$ released by the combustion of fossil fuels (*Raich & Potter, 1995*). Hence, slight shifts

in $R_s$ may cause profound changes in the atmospheric concentration of $CO_2$ and in the accumulation of soil carbon (*Schlesinger & Andrews, 2000*), thus subsequently affect global climate.

Considering the importance of forest ecosystems in the terrestrial carbon cycle and their response to global climate, $R_s$ and its dependence on environmental drivers have been the focus of numerous studies. For instance, soil temperature and moisture of soils are two of the major environmental drivers regulating $R_s$ (*Liu et al., 2016*). Additionally, disturbances, e.g., fire (*Muñoz Rojas et al., 2016*; *Köster et al., 2014*), harvesting (*Bahn et al., 2008*), artificial warming and precipitation changes (*Li et al., 2017a*), or land use changes (*Liu et al., 2011*; *Willaarts et al., 2016*) can also have large effects on $R_s$. $R_s$ is a complex biogeochemical process highly related to ecosystem productivity, leaf area index, and soil fertility (*Hibbard et al., 2005*), proving coupling between $CO_2$ assimilation by the vegetation and emissions from the soil (*Bahn et al., 2008*; *Hibbard et al., 2005*). $R_s$ is also influenced by the amount of litter (*Oishi et al., 2013*; *Wu et al., 2017*), vegetation type (*Mahecha et al., 2010*; *Wang et al., 2011*), and composition of the soil microbial community (*Luo et al., 2016*). However, many of the environmental drivers are correlated with each other, making it difficult to distinguish and quantify the contribution of each environmental factor.

Bamboo forests are widely distributed in warm temperate, subtropical and tropical zones between 46°N–47°S (*Lu et al., 2014*). Globally, bamboo forests cover 31.5 million ha (*FAO, 2010*). With more than 500 varieties and 39 species, China hosts the largest diversity of bamboo in the world, and the 6.16 million ha bamboo forests account for 2.97% of the total forest area in China (*SFAPRC, 2015*). Moso bamboo (*Phllostachys heterocycla cv. pubescens*) is appreciated for its rapid growth and high rate of timber production (*Guan et al., 2017*). Moso bamboo forest is a major forest type of subtropical forests in subtropical China (*Song et al., 2013*). Currently, the area covered by Moso bamboo forests increases annually by approximately 3%, mostly due to afforestation on wastelands (*Chen et al., 2009*), but also through conversion conifer and broadleaf forests and farmland (*Cui et al., 2011*; *SFAPRC, 2015*). Moso bamboo provides many benefits, including high income generation and other ecosystem services, to the forest owners.

Notably, the rate of carbon accumulation by Moso bamboo is high. Moso bamboo sequesters 4.91–5.45 t C ha$^{-1}$ each year (*Zhou & Jiang, 2004*), showing great potential for alleviating global warming by carbon fixation. Previous studies on Moso bamboo have concentrated on carbon storage, balance and its distribution in the ecosystem (*Li et al., 2013*), productivity of bamboo forest (*Cheng et al., 2015*; *Isagi et al., 1997*), and the variation in soil organic carbon stocks (*Guan et al., 2015*). Previous studies reported a close relationship between $R_s$ and biotic factors in other forest types (*Hibbard et al., 2005*), suggesting a coupling between forest canopy assimilation and carbon emissions from soil. However, comparatively little is known about bamboo forests. Thus, given the ecological importance of Moso bamboo forests at regional scale, there is a need for understanding the relationships between biotic and abiotic factors and $R_s$ in this kind of ecosystem.

In this study, we used soil respiration measurements from a Moso bamboo stand and combined these with measurements of abiotic and biotic factors. Our aims were to explore
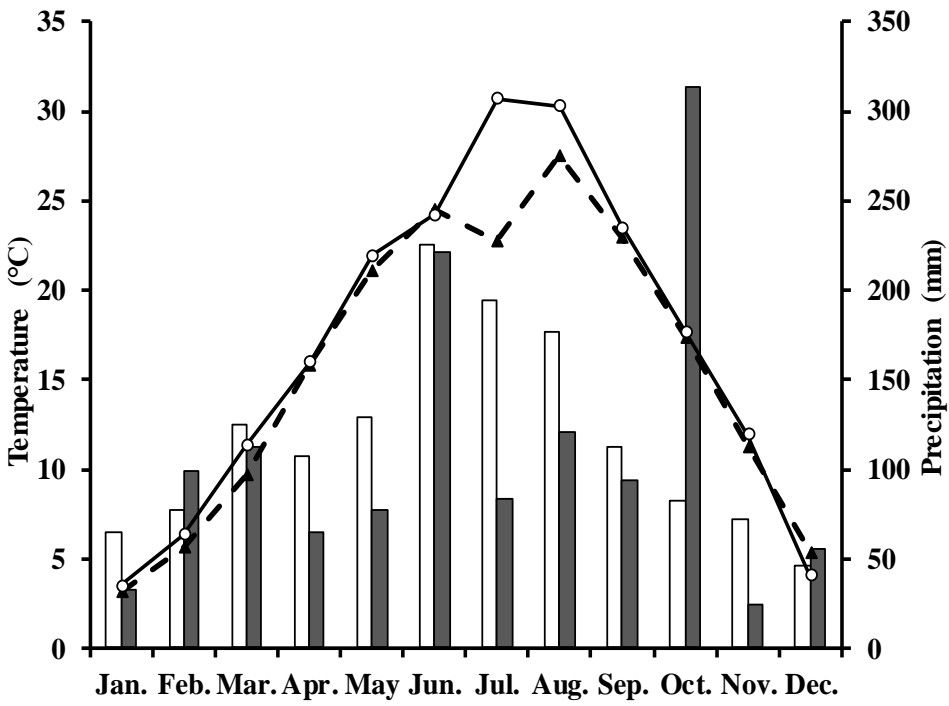

**Figure 1** **Monthly and long-term average air temperature ($T_a$) and precipitation at the study site.** White circles are monthly air temperature in 2013; black triangles denote long-term average air temperature; grey rectangles are monthly precipitation in 2013; white ones denote long-term average precipitation.

the temporal dynamics of soil respiration, and to identify the relative importance of the measured environmental factors.

## METHODS

### Study site

The measurements were done in a Moso bamboo forest with an eddy covariance flux tower in Anji, Zhejiang Province, southeast China (30°28′34.5″N, 119°40′25.7″E) at 380 m elevation. The study area has a typical subtropical monsoon climate with distinct seasons (*Li et al., 2018*; *Peel, Finlayson & McMahon, 2007*). The average annual air temperature and precipitation in 1981–2010 was 15.6 °C and 1,413.2 mm, respectively. Monthly average rainfall and air temperature in the study period are shown in Fig. 1. The soil type in this area is yellow red soil (Chinese system of soil classification), equivalent to Hapludult in USDA Soil Taxonomy (*Soil Survey Staff of United States, 1999*), pH is from 4.4 to 4.8, and soil bulk density is 1.5 g cm$^{-3}$ (*Chen, 2016*).

The study area (1km around the eddy covariance flux tower) was covered by 86.1% of Moso bamboo forest (*Xu et al., 2013*). The total area of the forest was 1,687 ha. Stand density was 3235 culms per hectare, the average canopy height was 11 m with a mean diameter at breast height of 9.3 cm. There was only a sparse understory in the study

area. The main management activities were harvesting 6- or 7-year old bamboos, and a proportion of new bamboo shoots each year. The forest was not fertilized nor weeded during the study period. Further detailed information of the site is found in *Mao et al. (2017)*. Moso bamboo has a biannual growth pattern. During "off years" (which are the even numbers in our site, i.e., 2012, 2014, 2016) few new bamboo shoots are produced, there is leaf senescence of old leaves, and new leaves grow vigorously (*Qiu, 1984*). During "on years," which are years with uneven numbers, more new bamboo shoots are produced and leaf senescence is limited. In our study site, the study period in 2013 was an "on-year".

## Experimental design and measurement
### Soil $CO_2$ flux measurement

The soil $CO_2$ flux was measured using an automated system consisting of a LI-8100 analyzer and a LI-8100-104 chamber and a multiplexer (LI-8150) (all LI-COR Inc., Lincoln, NE, USA). Soil respiration measurements were done at two hour intervals between 0:00 and 22:00 on selected sunny days for approximately two weeks (usually from day 10 to day 23 of every month) of every month in 2013. The duration of each flux measurement was 2 min and the fluxes were calculated by an exponential fit of $CO_2$ against time by Soil Flux Pro, version. One 40 m × 40 m plot was established around the flux tower within the forest. Sixteen sampling polyvinyl chloride (PVC) soil collars (20 cm inside diameter, 10 cm height, and 5 cm plugged into the soil) were randomly placed within the plot. All collars remained permanently in place throughout the study period. There were few herbs in the Moso bamboo forest. To reduce the disturbance-induced carbon dioxide emission, the first measurement started at least 24 h after insertion. The areas inside collars were kept free of plants by cutting the plants carefully using scissors about monthly during the year. The data and the performance of the equipment were checked regularly to ensure the reliability of measurements throughout the year. Soil water content (SWC, $m^3\ m^{-3}$) and soil temperature ($T_s$, °C) were monitored adjacent to each collar at 5 cm depth with 2 theta probes inserted vertically (ML2x; Delta-T Inc., Cambridge, UK; Omega Inc., Norwalk, CT, USA) provided with the system. We defined March to May as spring, June to August as summer, September to November as autumn, and January, February and December as winter.

### Measurements of environmental variables at the eddy covariance site

$T_s$ and SWC were monitored by soil temperature sensors (109SS, Campbell Inc.) and soil moisture sensors (CS616; Campbell Inc., Logan, UT, USA), respectively, at 5 cm, 50 cm and 100 cm depths ($T_{s5}$, $T_{s50}$, $T_{s100}$, $SWC_5$, $SWC_{50}$, $SWC_{100}$) close to the eddy covariance tower. Air temperature and relative humidity were measured using HMP45C probes (Vaisala, Helsinki, Finland) at 1 m, 7 m, 11 m, 17 m, 23 m, 30 m, and 38 m above the ground. All the data were recorded by a data logger (CR1000; Campbell Inc., USA) and saved as 30-min averages.

### Biological factors measurements

Gross ecosystem productivity (GEP) was obtained by eddy covariance (EC) technique. An open-path infrared gas analyzer LI-7500 (Li-Cor Inc., Lincoln, NE, USA), in conjunction
with a 3-dimensional sonic anemometer CSAT3 (Campbell Inc., Logan, UT, USA), was placed at 38 m above the ground. All the raw flux data were sampled at 10 Hz, and calculated and recorded by a CR1000 data logger (Campbell Inc., USA) as 30-min average values. The flux data was processed using the EdiRe software (University of Edinburgh). A double-coordinate rotation was applied and the Webb-Pearman-Leuning correction was conducted to remove the effects of air-density fluctuations. Daily net ecosystem exchange (NEE) was calculated as the daily sum of the measured $CO_2$ flux and the daily rate of change in $CO_2$ storage below the height of the EC system. Ecosystem respiration (RE) was calculated for each 30-min by extrapolating the exponential regressions between the night NEE at high-friction velocity and soil temperature at the 5 cm depth and summed into the daily values. Daily gross ecosystem productivity (GEP) was estimated as the difference of daily RE and daily NEE (*Song et al., 2017*).

The flux data were discarded when the following errors were observed (*Yan et al., 2013*; *Yu et al., 2006*; *Song et al., 2017*): (1) the $CO_2$ flux was beyond the range of $-2.0$ to $2.0$ mg $CO_2$ m$^{-2}$ s$^{-1}$, $CO_2$ concentration was $< 500$ or $>800$ mg m$^{-3}$, and water vapor concentration was outside the range of $0$–$40$ g m$^{-3}$; (2) abnormal values, i.e., when the absolute value of the difference between a numerical value and a continuous five points was $>2.5$ times of its variance; (3) the measurements occurred during precipitation events; (4) the number of valid samples was $< 15,000$; (5) friction velocity was low (u* $< 0.2$ m s$^{-1}$). Gaps occurred more frequently at night than during the day. After data filtering, the annual flux data 64% of the data were retained.

Gaps less than 2 h were linearly interpolated, gaps more than 2 h were filled with the look-up-table method, which were built up based on the two-adjacent-month periods and two main environmental factors (photosynthetically active radiation and air temperature). For details information, please see the literature by *Song et al. (2017)*.

Leaf area index (LAI) was measured at 6:00–10:00 and 15:00–17:50 of sunny, no cumulus days and with good visibility days. Measurements were done monthly using digital camera provided with a fish-eye lens in combination with MODIS LAI following the methods of *Li et al. (2017b)*. LAI was reported as the average of three sample points chosen within the 20 m $\times$ 20 m plot on non-rainy days. The LAI data was calculated as mean values $\pm$ SD (standard deviation).

## Data analysis

We analyzed the soil respiration as a function of soil temperature assuming an exponential $Q_{10}$ type relationship.

$$R_s = ae^{bt} \tag{1}$$

$$Q_{10} = e^{10b} \tag{2}$$

where $R_s$ ($\mu$mol m$^{-2}$ s$^{-1}$) is soil respiration, $T$ is soil temperate at 5 cm depth, $a$ and $b$ are fit parameters, Eq. (1) (*Van't Hoff, 1884*). The temperature sensitivity parameter, $Q_{10}$, was calculated by Eq. (2) (*Sheng et al., 2010*; *Song et al., 2013*).

One-way analysis of variance (ANOVA) and the least significant difference were carried out to test the statistical significance of differences in soil respiration, environmental

**Table 1** Relationships between soil respiration ($R_s$) and soil temperature measured by Li-8150 ($T_s$) in 2013.

| Time | Equation | $R^2$ | $Q_{10}$ | F | P |
|------|----------|-------|----------|---|---|
| Dec.~Feb. | $R_s = 0.279\exp(0.241*T_s)$ | 0.684 | 11.08 | 73.74 | 0.000 |
| Mar.~May | $R_s = 0.629\exp(0.095*T_s)$ | 0.819 | 2.59 | 154.39 | 0.000 |
| Jun.~Aug. | $R_s = 1.427\exp(0.058*T_s)$ | 0.627 | 1.79 | 57.08 | 0.000 |
| Sep.~Nov. | $R_s = 0.594\exp(0.107*T_s)$ | 0.983 | 2.92 | 1976.33 | 0.000 |

**Notes.**

$R_s$, soil respiration; $T_s$, soil temperature measured by Li-8150.

**Table 2** Correlation coefficients of monthly mean soil $CO_2$ fluxes and its affecting factors in 2013.

| Factors | $R_s$ | Environmental variables | | | | | | | GEP |
|---------|-------|-------|--------|---------|-------|-----|---------|----------|-----|
| | | $T_s$ | $T_{s5}$ | $T_{s50}$ | $T_a$ | Rh | $SWC_5$ | $SWC_{50}$ | |
| $T_s$ | 0.988** | | | | | | | | |
| $T_{s5}$ | 0.968** | 0.99** | | | | | | | |
| $T_{s50}$ | 0.966** | 0.95** | 0.97** | | | | | | |
| $T_a$ | 0.966** | 0.99** | 0.99** | 0.946** | | | | | |
| Rh | 0.21 | 0.21 | 0.152 | 0.133 | 0.081 | | | | |
| $SWC_5$ | −0.229 | −0.135 | −0.153 | −0.348 | 0.337 | 0.438 | | | |
| $SWC_{50}$ | 0.244 | 0.306 | 0.296 | 0.142 | 0.334 | 0.688* | 0.813* | | |
| GEP | 0.841** | 0.868** | 0.863** | 0.752* | 0.894** | 0.198 | 0.148 | 0.555 | |
| LAI | 0.937** | 0.89** | 0.91** | 0.914** | 0.901** | 0.15 | −0.275 | 0.162 | 0.761* |

**Notes.**

$T_s$ (soil temperature measured by Li-8150 probe), Rh (air relative humidity measured by flux tower at 1m height), GEP (gross ecosystem productivity), other variables shown see Fig. 2. Statistical significance with:

** $p$-values <0.01, * $p$-values < 0.05; besides, due to no significant correlation between soil moisture and other factors, it was not shown in Table 1 (expect GEP in July and August).

(Table 1) and biotic factors (Table 2) between seasons. Regression (including nonlinear and linear regression) and correlation analysis was performed to analyze the relationship between soil respiration, biotic and abiotic variables. All analyses were conducted using the PASW software (PASW Statistics 18.0 for windows, SPSS Inc., Chicago, IL, USA).

## RESULTS

### Seasonal dynamics of environmental and biotic factors in Moso bamboo forest

In 2013, the annual average air temperature was 1.2 °C higher and total precipitation 114.5 mm lower than the long-term averages. The 30.7 °C in July and 30.3 °C in August (Fig. 1) were as much as 7.9 and 2.8 °C higher, respectively, than the long-term averages. Precipitation was 57.2% and 31.5% of the long-term average in July and August, respectively. The annual rainfall in 2013 was 1,298.7 mm, and occurred mostly from May to October. Additionally, it decreased by 57.18% in July compared with the corresponding period of long term (Fig. 1), showing exceptionally hot and dry conditions. Temperatures at different soil depths ($T_{s5}$, $T_{s50}$) and air temperature at 1m height ($T_a$) exhibit a similar seasonal pattern (Fig. 2A): a gradual increase from January to July, maximum in July, and

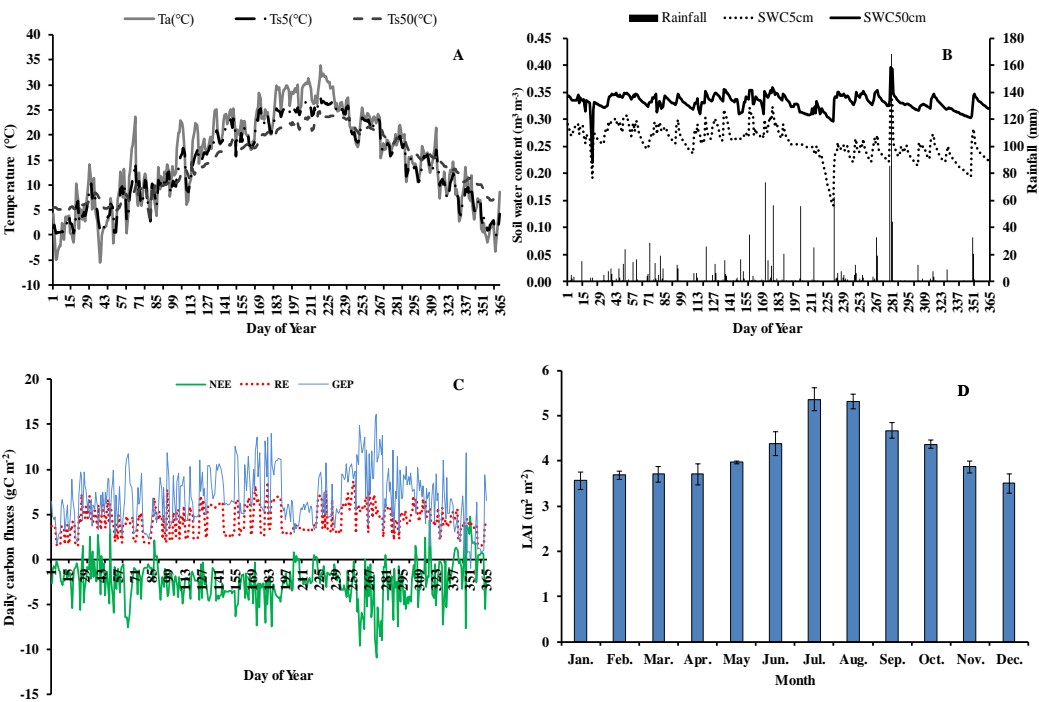

**Figure 2** **Seasonal variation of abiotic and biotic factors of Moso bamboo forest in 2013.** (A) daily temperature (°C) of air (Ta) and soil at 5 cm ($T_{s5}$), 50 cm ($T_{s50}$) depth. (B) Daily rainfall amount (mm) and soil water content (m³ m⁻³) at 5 cm depth (SWC₅) and 50 cm depth (SWC₅₀). (C) Daily carbon fluxes (NEE, RE, GEP, gC m⁻²). (D) Mean monthly LAI (m² m⁻²) during the study period Mean ±SD ($n = 3$).

a slow decrease till December. $T_{s5}$ and $T_{s50}$ changed comparatively more smoothly and steadily than $T_a$. Soil water content at 5 cm and 50 cm depths (SWC₅ and SWC₅₀) were obviously affected by rainfall, and were at the lowest in July and August.

Seasonal variation in net ecosystem exchange (NEE), ecosystem respiration (RE) and gross ecosystem productivity (GEP) showed several peaks during 2013. The lowest mean daily NEE was detected in August (0.76 g C m⁻²) (Fig. 2C), and highest in June and September. Additionally, NEE was positive on some rainy and cloudy days. Mean daily NEE, RE and GEP was −2.11 g C m⁻² day⁻¹, 5.36 g C m⁻² day⁻¹ and 7.48 g C m⁻² day⁻¹, respectively. Due to the impact of drought, GEP decreased significantly in July and August, being 59.9% and 80.0%, respectively, of GEP in the corresponding period in 2011 (*Chen, 2016*). LAI remained at approximately 3.6 in winter and spring, increased gradually starting from March, and reached a maximum (5.92) in July (Fig. 2D). Thereafter, LAI decreased slowly, exhibiting the typical growth characteristic of Moso bamboo in an "on year" (*Chen, 2016*).

## Diurnal variation of soil CO₂ fluxes and its response to temperature

Soil respiration ($R_s$) in our forest presented similar diurnal dynamics across all seasons (Fig. 3A). After a daily minimum occurring between 05:00 to 07:00, it increased slowly reaching the maximum value between 14:00 to 16:00, and then decreased gradually. There
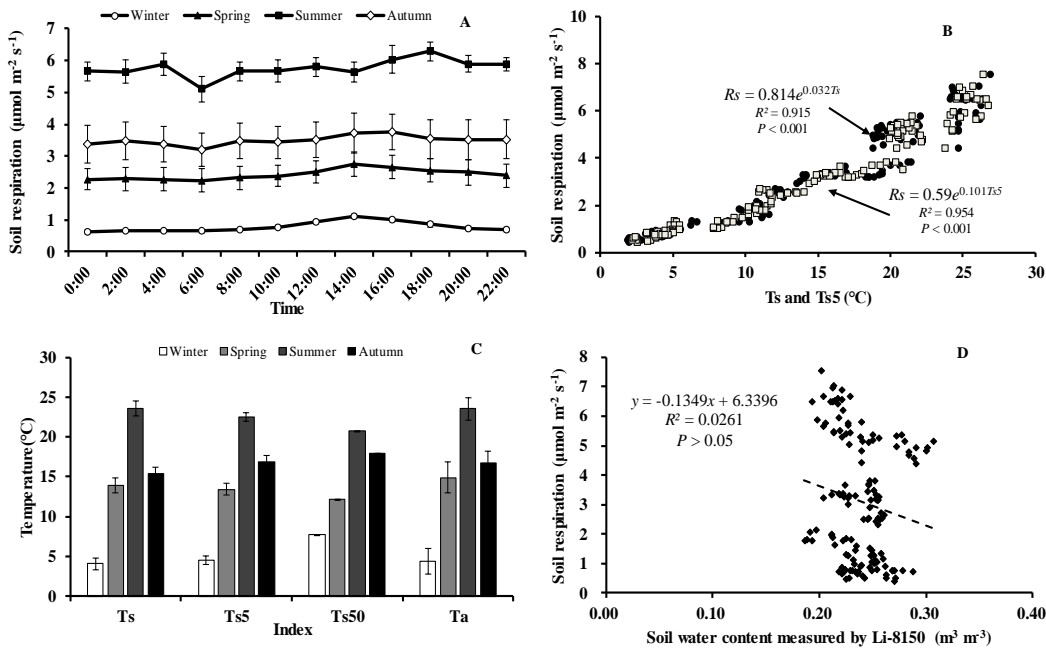

**Figure 3** **Diurnal, seasonal dynamic of soil respiration and the relationship between related factors and soil respiration in Moso bamboo forest.** (A) error bars denote standard error of means ($n = 12$). Seasonal variation of soil respiration (B) $T_s$, black circle, $T_{s5}$, white diamond ($n = 144$), and (C) seasonal variation of different temperatures; (D) relationship between soil water content and soil respiration ($n = 144$) error bars indicate standard deviation of the means ($n = 12$).

were, however, big differences in $R_s$ between months. Monthly maximum values of $R_s$ ranged from 0.75 in January to 7.52 μmol m$^{-2}$ s$^{-1}$ in August.

Monthly mean values of $R_s$ correlated positively with both soil temperature at 5 cm depth ($T_{s5}$) measured by the EC system and air temperature ($T_a$) ($P < 0.01$, not shown), with the correlation with $T_{s5}$ being higher (Fig. 3B, Table 2). An exponential relationship was used to estimate $R_s$ based on $T_s$ (Table 1). $T_s$ explained 69.7% variation of the variation in $R_s$ at a diurnal scale, whereas $T_{s5}$ explained 63.9% (not shown). Both exponential regression models were statistically significant ($P < 0.01$). Plotting the diurnal variation of $R_s$ against $T_s$, and $T_{s5}$ (Fig. 4) showed a clear hysteresis. Additionally, there was slight discrepancy in the elliptic shape of $T_s$ and $T_{s5}$, and the subtle difference in elliptic shape of both could explain the coefficient or determination ($R^2$) of exponential regression in the relationship of $T_s$ and $T_{s5}$ (not shown).

## Seasonal dynamics of soil CO$_2$ fluxes and its driving factors

Soil respiration followed a clear seasonal pattern in soil respiration (Fig. 3A), being highest in summer with 5.77 μmol CO$_2$ m$^{-2}$ s$^{-1}$, followed by autumn (3.50 μmol CO$_2$ m$^{-2}$ s$^{-1}$), and spring (2.42 μmol CO$_2$ m$^{-2}$ s$^{-1}$), and lowest in winter (0.76 μmol CO$_2$ m$^{-2}$ s$^{-1}$). The average annual soil CO$_2$ flux was 3.11 μmol CO$_2$ m$^{-2}$ s$^{-1}$, equating to an annual $R_s$ of 44.07 t CO$_2$ ha$^{-1}$ a$^{-1}$. Temperatures at different heights and depths presented similar

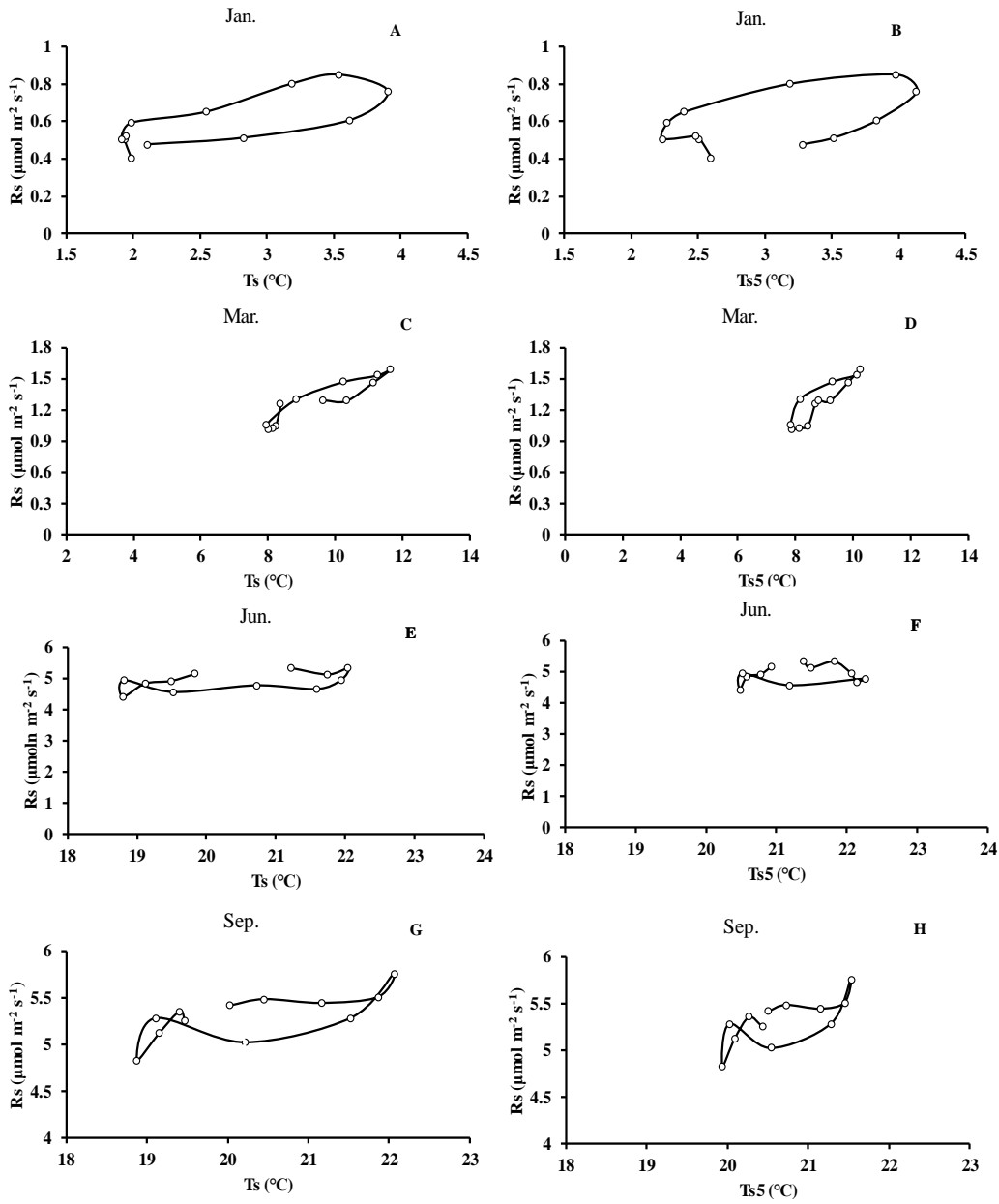

**Figure 4** **Mean diurnal changes of $R_s$ in response to $T_s$ and $T_{s5}$ in different months of Moso bamboo forest.** $R_s$ denotes soil respiration; $T_s$ denotes soil temperature measured by Li-8150; $T_{s5}$ denotes soil temperature at 5 cm depth measured by eddy covariance technique. One month of each season was chosen.

seasonal dynamics, being highest in summer and lowest in winter (Fig. 3C). Furthermore, $Q_{10}$ values were small in summer and large in winter (Table 1).

Monthly mean values of LAI, soil temperature and GEP were all significantly related to soil respiration (Table 2 and Fig. 5).

Within each seasonal, there was a complex linear relationship between SWC and $R_s$, with significant ($P < 0.01$) negative correlation in summer ($R = -0.796$, $R_s = -19.101{*}SWC +$

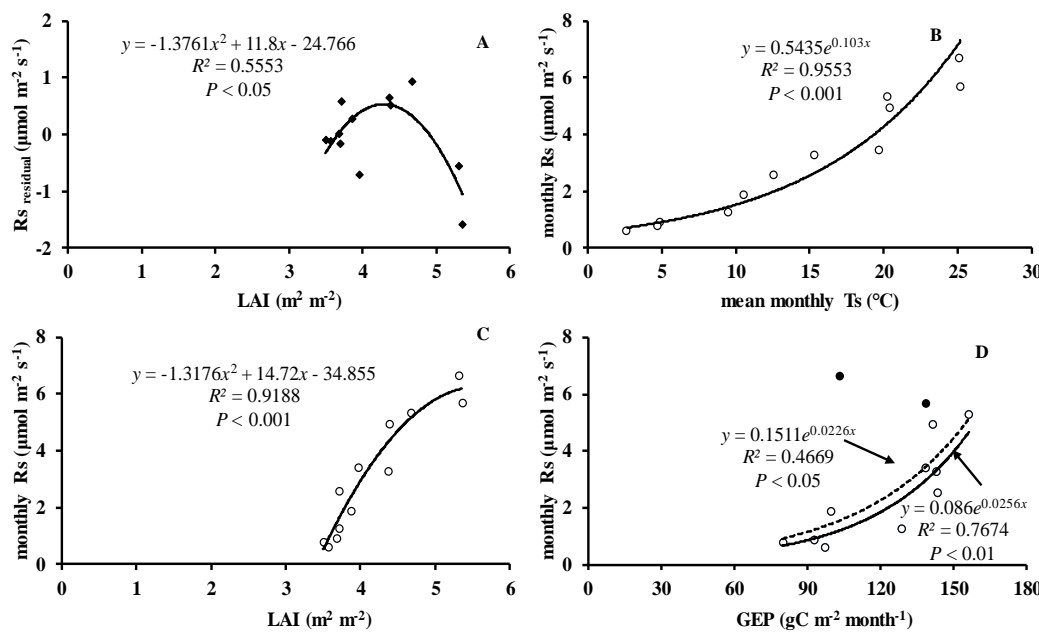

**Figure 5** **Relationship between monthly soil respiration and leaf area index, gross ecosystem productivity.** (A) Residuals of observed minus predicted (calculated by the best model in the last row of Table 3) values of $R_s$ in relation to monthly values of LAI. Monthly $R_s$ in relation to (B) mean monthly soil temperature, (C) LAI, and (D) monthly GEP. Black circles denote GEP of July and August, hollow circles are months excluding July and August; dotted line is relationship including all the months in 2013, whereas solid line is excluding GEP of July and August.

**Table 3** **Relationship between $R_s$, $T_s$ and $SWC$.** Coefficients of determination ($R^2$) and root mean square error ($RMSE$) were given.

| Model | References | $R^2$ | a | b | c | d | RMSE |
|---|---|---|---|---|---|---|---|
| $R_s = \exp(a + b*T_s)*SWC$ | Gao, Guo & Liu (2011) | 0.895 | 1.07 | 0.09 | – | – | 0.663 |
| $R_s = (c*SWC + d)*a*\exp(b*T_s)$ | Han, Zhou & Xu (2008) | 0.918 | 0.64 | 0.08 | 1.13 | 0.97 | 0.591 |
| $R_s = \exp(a + b*T_s + c*SWC + d*T_s*SWC)$ | Li et al. (2000) | 0.919 | 0.22 | 0.05 | −1.97 | 0.14 | 0.588 |
| $R_s = \exp(a + b*T_s + c*SWC + d*SWC^2)$ | Tang & Baldocchi (2005) | 0.922 | 1.88 | 0.08 | −18 | 39 | 0.578 |
| $R_s = a + b*T_s + c*SWC + d*T_s*SWC$ | Wang et al. (2003) | 0.929 | −3.74 | 0.47 | 13.45 | −0.9 | 0.542 |
| $R_s = a + b*\exp(c*T_s) + d*T_s*SWC$ | Zhou et al. (2008) | 0.936 | −4.73 | 4.76 | 0.03 | −0.04 | 0.515 |

**Notes.**
The abbreviation was shown in Fig. 1. $P$ value of every model was 0.000.

10.368), positive linear correlational in autumn ($P < 0.01$, $R = 0.552$, $R_s = 47.663*SWC-7.012$) and spring ($P<0.05$, $R = 0.331$, $R_s = 36.661*SWC-6.708$), but no correlation ($P > 0.05$) in winter ($R = 0.008$), indicating that SWC played crucial role in $R_s$ at the growing period of Moso bamboo. Soil temperature and soil moisture showed significant linear relationship in the summer ($R = −0.939$, $T_s = −0.013*SWC + 0.559$, $P < 0.001$).

An exponential equation model was used to fit the relationship between different temperatures ($T_s$, $T_{s5}$) and soil respiration (Fig. 3C). The equations of $T_{s5}$-$R_s$ ($R^2 = 0.954$) and $T_{s50}$- $R_s$ ($R^2 = 0.929$) both showed higher $R^2$ than that of $T_s$- $R_s$ ($R^2 = 0.915$), possible because of the relative stability of soil temperature profile measurement in eddy covariance

system. Furthermore, due to the complex relationship between SWC and $R_s$, as well as considering combination of temperature and soil moisture, six models were compared that predict $R_s$ based on soil temperature and soil moistures (Table 3). Based on *RMSE* and $R^2$, the model ($R_s = a + b * \exp(c * T_s) + d * T_s * SWC$) showed best result, suggesting $T_s$ and SWC could explain 93.6% temporal variation of $R_s$ in 2013. Compared with a soil temperature($T_s$)-soil respiration($R_s$) equation (Fig. 3C, $R^2 = 0.915$), It showed a slight increase $R^2$ (Table 3, $R^2 = 0.936$).

## DISCUSSION

Our work demonstrates the importance of three factors that affect soil respiration in Moso bamboo: temperature, soil water content and either productivity or LAI. The importance and interactions of the factors will be discussed subsequently.

Of the three factors, soil temperature was the dominant driver of soil respiration with an $R^2$ of over 0.8 (Figs. 3C and 5B). Seasonal change of $R_s$ has been investigated in varying ecosystems. Soil temperature and soil water content are commonly considered to be two major determinants of seasonal variations in measured $R_s$ (*Davidson, Belk & Boone, 1998*; *Davidson, Janssens & Luo, 2006*; *Davidson et al., 2012*; *Sihi et al., 2018*). In this study, soil respiration increased with the rising of soil temperature. Similar results were explored by *Shi, Wang & Liu (2012)* on a global scale. However, soil temperature explained only 62.7% variation of soil respiration during summer (June, July and August). This was not only due to a lower variation of soil temperature during summer months, but also, as shown in Table 1, the temperature sensitivity of soil respiration was markedly lower in the summer, which was likely caused by low SWC values. Additionally, plots of soil respiration against daily temperature patterns show a rather flat relationship for the summer with a strong hysteresis. Similar findings have been reported in Moso bamboo forest of subtropical China by *Tang et al. (2016)* and *Song et al. (2013)*. Depth of the soil temperature measurement affected the explanatory power of soil temperature. The explanatory power of the temperature in the organic layer was highest and decreased with the depth of the measurements. This indicates that most of the respiration originates from the organic layer (*Davidson et al., 2006*). *Zhang et al. (2016)* made similar observations in winter wheat ecosystems. While *Dai et al. (2004)* found soil respiration of wheat was highly correlated with soil temperature at 10 cm depth.

The relationship between soil carbon efflux and soil temperature showed a diurnal hysteresis (Fig. 4). This indicates that there is a delayed effect of the rapidly varying temperature and diurnal variation of soil respiration, similar to the studies by *Högberg et al. (2008)*, *Abramoff, Davidson & Finzi (2017)* and *Savage et al. (2009)*. One explanation is different diurnal temperature pattern at different depths and delays due to the transport of $CO_2$ from the sites of respiration to the soil surface (*Graf et al., 2008*). Furthermore, other research suggested that the length of the delay could vary among different species (*Raich & Schlesinger, 1992*). The hysteresis could also be an artifact for measuring soil temperature at a different depth than respiration is occurring. However, we tried different depths to measure soil temperature. Since the depth of the measurements of soil temperature varies between studies, it might be difficult to compare the sensitivity of soil respiration

to soil temperature between studies (*Zhang et al., 2016*). Previous research suggested diurnal variation of $R_s$ was out of phase with corresponding $T_s$ at 2 cm depth, resulting in significant hysteresis (*Gaumont-Guay et al., 2006*). As discussed above, there may be two possible reasons: (1) effects of diurnal variations of root respiration supplied by newly produced photosynthetic products and (2) diurnal variations of soil water content near the critical value (*Bahn et al., 2008*; *Davidson et al., 2012*; *Wang et al., 2015*; *Sihi et al., 2018*), while further reasons for this (especially in winter) are needed controlled experiments to explore and demonstrate.

The relationship between soil respiration and soil moisture was more complicated in our study. Soil moisture improved marginally our models of soil respiration with a better fit of the models particularly in the dry summer 2013. No significant correlation was found between soil respiration and soil moisture in 2013 (Fig. 3D). Similar findings had been reported for Moso bamboo forest in Zhejiang province (*Song et al., 2013*). However, soil moisture had a negative statistically significant ($P < 0.001$, $R = -0.796$, $R_s = -19.101*SWC +10.368$) correlation with soil respiration in summer while correlation in the other seasons was positive. Previous observation indicated a pronounced correlation between $R_s$ and SWC in subtropical forests (*Sheng et al., 2010*; *Liu et al., 2011*). The negative correlation of soil respiration and soil moisture in our study was probably caused by a spurious correlation of soil temperature and soil moisture during summer ($R = -0.939$, $T_s = -0.013*SWC +0.559$, $P < 0.001$). The cause of a nonexistent or negative linear correlation between SWC and Rs could be that natural variation of SWC covers only a part of response curve (at low to medium SWC, $R_s$ depends positively on it because water is limiting, then there might be a plateau and at high SWC oxygen transport to the soil depth and transport of $CO_2$ back might be blocked) (*Linn & Doran, 1984*; *Xu, Baldocchi & Tang, 2004*). When we fitted non-linear models to soil respiration using temperature and soil moisture we got only a small increase in the $R^2$ when soil moisture was included into the model. We have also checked the interaction between SWC and temperature (shown in Table 3) and our best model (last row in Table 3) shows that a model which includes interactions was the best. This indicates that soil moisture was, even in the dry year of 2013, not an important limitation of soil respiration.

The models of soil respiration suggest that the temperature sensitivity of soil respiration declines when soil moisture is decreasing (*Almagro et al., 2009*; *Jassal et al., 2008*; *Wang et al., 2006*), this may be due to the diurnal variation of soil moisture near the criticality value. Also, $Q_{10}$ varied over the different seasons (Table 1). Due to smaller amplitude of soil temperature in deeper layers (*Pavelka et al., 2007*), $Q_{10}$ values estimated from deeper soil layers tended to be larger than those of shallower layers. This can partly explain the discrepancy between $T_a$, $T_s$, and $T_{s5}$. $Q_{10}$ was about 2.80 in our study, within range of 1.33~5.53 estimated for forests in China (*Chen et al., 2008*), lower than 4.09 in Moso bamboo forest of central Taiwan (*Hsieh et al., 2016*), but higher than median of 2.0~2.4 (*Hashimoto, 2005*).

Previous observation pointed out that annual $Q_{10}$ value was not only an indicator of the response to soil temperature, but also a comprehensive response to variations of

other factors (i.e., SWC, root biomass, root growth, amplitude of $R_s$, and other seasonal processes, (*Yuste, Janssens & Carrara, 2004*)).

Another driver of soil respiration is the phenology of Moso bamboo which shows a large variation in below ground activities. In the spring, carbon is allocated to the production of new bamboo shoots. After bamboo has completed its main growth period in summer and new leaves are fully-expanded, it accumulates nutrient substance and allocates its main growth to the rhizome. Then in autumn Moso bamboo starts to hatch bamboo shoots for the next year (*Chen, 2016*). In this growing phase, soil moisture was a key factor for soil respiration. Subsequently, the stand got into overwintering stage. Soil moisture became less important in this period. Consequently, the importance of soil moisture for soil respiration varies among seasons and was more important during the time of active growth of Moso bamboo. However, soil temperature rather than soil moisture remained the most important drivers of soil respiration (*Janssens & Pilegaard, 2003*).

The explanation for the differences in soil respiration and $Q_{10}$ values are driven by the seasonal pattern of gross primary production which drives substrate supply to the root and rhizosphere (*Bahn et al., 2008*). Currently several authors have reported productivity should be considered to improve the prediction of soil respiration (*Bahn et al., 2008*; *Hibbard et al., 2005*; *Vargas et al., 2011*; *Zhang et al., 2016*). Numerous studies have shown close relations between soil respiration and canopy photosynthesis at different timescales. *Högberg et al. (2008)* reported that soil respiration was largely driven by recent primary production of the vegetation. Monthly soil respiration was significantly related to LAI and GEP in our study (Figs. 5A, 5C and 5D). The finding agreed with the view of a coupling of photosynthesis and soil respiration. Likewise, *Yuste, Janssens & Carrara (2004)* found that seasonal $R_s$ was positively related to LAI. *Bahn et al. (2008)* suggested $R_s$ was closely related to LAI across grassland sites. In our study, LAI was closely related to the productivity of vegetation. There was similar monthly variation pattern of LAI and $T_a$ in our study, which in turn increased the difficulty to detect relationships of $R_s$ in relation to biological variable. Soil respiration is a complex biological process, composed of several processes from both autotrophic and heterotrophic organisms. Besides soil temperature and soil water content, it is known that soil respiration is partly explained by forest type, stand age and altitude in subtropical forests (*Wang et al., 2011*). Additionally, other variables such as management (i.e., fertilization, thinning and harvesting activities, *Gao et al., 2014*; *Liu et al., 2011*), litter, soil microbial (*Linn & Doran, 1984*) and physical properties, root biomass and extreme weather (e.g., warming, precipitation events, short-term drought events), all have indirect and direct effects on soil respiration. However, how these influence autotrophic and heterotrophic processes is not well understood and should be a subject of further research.

## CONCLUSIONS

Soil respiration ($R_s$) in the Moso bamboo forest exhibited both daily and seasonal dynamic patterns, with its highest values in summer and lowest values in winter. Soil respiration correlated positively with soil temperature ($P < 0.01$), which explained 69.7% of variation in $R_s$ at a diurnal scale and 91.5% of variation in $R_s$ for the whole year. $R_s$ correlated

positively with soil moisture in spring, autumn, and negatively in summer, implying that moisture played a crucial role in different growth phases, but did not correlate significantly on daily scale, this may result from soil carbon substrate supply limiting soil microbial respiration in summer, and enhancing soil respiration in winter given substrate diffusion to the reaction site, which is generally driven by the thickness of the soil water film. The model that included soil temperature and soil moisture explained 93.6% of the seasonal variation in $R_s$. The relationship between $R_s$ and different soil temperatures exhibited a clear hysteresis. Soil respiration correlated positively ($P < 0.01$) with gross ecosystem productivity and LAI in our study, showing the significance of biotic factors in affecting soil respiration, and a need for future research to analyze the relationship between canopy photosynthesis and soil $CO_2$ flux.

## ACKNOWLEDGEMENTS

We thank Li Y for giving comments that greatly improved this manuscript. Tan Y provided field assistance. And we also would like to thank Professor Jiang Hong's team for their invaluable support with the flux data collection. The authors also would like to thank the editor, three reviewers and Prof. Petri Penttinen for their contribution to the peer reviews of this study.

### Funding

This study was supported by the National Natural Science Foundation of China (31670644, 31370637), Natural Science Foundation of Zhejiang Province (LR14C160001), 973 Program of China (2011CB302705), Fund of State Key Laboratory of Subtropical Silviculture (ZY20180201), Joint Research fund of Department of Forestry of Zhejiang Province and Chinese Academy of Forestry (No. 2017SY04), Zhejiang Provincial Collaborative Innovation Center for Bamboo Resources and High-efficiency Utilization, the Key Science and Technology Projects of Zhejiang Province (2015C03008), and the Key Discipline of Forestry of Creative Technology Project of Zhejiang Province (201510). The funders had no role in study design, data collection and analysis, decision to publish, or preparation of the manuscript.

### Grant Disclosures

The following grant information was disclosed by the authors:
National Natural Science Foundation of China: 31670644, 31370637.
Natural Science Foundation of Zhejiang Province: LR14C160001.
973 Program of China: 2011CB302705.
Fund of State Key Laboratory of Subtropical Silviculture: ZY20180201.
Joint Research fund of Department of Forestry of Zhejiang Province and Chinese Academy of Forestry: 2017SY04.
Zhejiang Provincial Collaborative Innovation Center.

Key Science and Technology Projects of Zhejiang Province: 2015C03008.
Key Discipline of Forestry of Creative Technology Project of Zhejiang Province: 201510.

## Competing Interests

Frank Berninger is an Academic Editor for PeerJ. The authors declare there are no competing interests.

## Author Contributions

- Yuli Liu, Guomo Zhou and Huaqiang Du conceived and designed the experiments, performed the experiments, analyzed the data, contributed reagents/materials/analysis tools, prepared figures and/or tables, authored or reviewed drafts of the paper, approved the final draft.
- Frank Berninger analyzed the data, contributed reagents/materials/analysis tools, prepared figures and/or tables, authored or reviewed drafts of the paper, approved the final draft.
- Fangjie Mao performed the experiments, analyzed the data, contributed reagents/-materials/analysis tools, authored or reviewed drafts of the paper, approved the final draft.
- Xuejian Li and Liang Chen performed the experiments, analyzed the data, contributed reagents/materials/analysis tools, prepared figures and/or tables, approved the final draft.
- Lu Cui, Yangguang Li and Di'en Zhu performed the experiments, analyzed the data, approved the final draft.

## Data Availability

The raw data are provided in a Supplemental File.

## Supplemental Information

Supplemental information for this article can be found online at http://dx.doi.org/10.7717/peerj.5747#supplemental-information.

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
