# Peer review of "Soil respiration of a Moso bamboo forest significantly affected by gross ecosystem productivity and leaf area index in an extreme drought event"

_PeerJ, doi:10.7717/peerj.5747_

## Round 0.1 · original submission · Major Revisions

Although one of the reviewers recommends that the manuscript be rejected, I think the authors can address the comments. However, those comments are extensive. Also please note that reviewer 1 included an annotated manuscript with comments.

·

Basic reporting

Please see the "General comments for the author" section and the attachment.

Experimental design

This is a primary research article and it is within the Aim and Scope of the journal. The question addressed in this piece is relevant, which is based on knowledge gap on soil respiration and its drivers in Moso bamboo forest. Overall, the experimental design undertaken for this study is standard and sufficient details have been provided to ensure replication.

Validity of the findings

A better understanding of the biotic and abiotic drivers of soil respiration is prerequisite for estimating ecosystem carbon budget. To that end, the findings in this article are valid and one of its first kind as the study was conducted in one of the important forest ecosystems of China.

Additional comments

Liu et al., evaluated the dynamics of soil respiration within a tower footprint in a Moso bamboo forest of SE China. Overall, the results are straight forward. However, some work is still needed to facilitate a better flow of the story. Additionally, authors need to pay attention to grammatical errors (and terminology) before final submission. See the attached pdf for more details.
Also appreciate sharing the raw data. However, adding a README file will be useful for a short description of the columns (and units) related to each variable.

Reviewer 2 ·

Basic reporting

The language quality starts very promising, but unfortunately drops at several points, most noteworthy the Materials and Methods section - almost looks as if someone else has written (or strongly edited) different parts. As a non-native speaker, I can only give a few examples (see general comments). The paper should be thoroughly checked, e.g. by whoever checked the introduction.

I am not familar with the (interesting) concept of asking submitters to provide raw data and asking referees explicitly to judge from them, and from the presented figures and tables, if there was any inappropriate manipulation. Of course I would report signs of manipulation if stumbling accross them, but I don't think that most (especially "well-done") types of manipulation can be excluded this way. I am unsure how strict the request to submit raw data should be interpreted, but maybe I should mention that the data submitted by the authors are not "raw data" in the strict sense. The closest match to "raw data" for this study would be the so-called "*.81x" files produced by the software of the chamber manufacturer (Li-Cor). They require a special (but free) software to conveniently look at, but can also be read with a simple text editor and would make undetected manipulation harder.

Experimental design

The design has all ingredients for a comprehensive successful study focused on but not limited to soil respiration, especially given the fact that LAI and eddy-covariance measurements of NEE are at hand. However, the potential of these extra data seems only partly exploited.

One point that requires a clearer description in the methods section and more presentation of different raw/aggregated states of the results is that despite using an automated chamber system, measurements were apparently performed only on selected days. Please get clearer about the reasons, the strategy resulting from it (see also general comments) and the resulting number of available raw data points.

Validity of the findings

The presented results look sound (apart from the fact that it would be good to also show some not-yet-aggregated diurnal cycyles), but the discussion is sometimes speculative and sometimes superficial.

Additional comments

L 58: Maybe consider replacing the "kingdom" by a more objective description, e.g. if China has the largest bamboo area or production, mention this.

L 62-64: It would be interesting to briefly report which former ecosystems or land use types are replaced by this growth.

L84-87: Sentence unclear (why "As.."/where is the verb?). Also it is unclear which data are from measurements and where the classification comes in. Give reference for the classification if possible.

L88: This number makes no sense without telling how the accumulation was done, e.g. adding mean temperatures of each day above 10°C? Then technically the unit is °C*day, not °C. If this is a common technique give the reference.

L89: Language example: are (here), not were, because the showing happens now. Similarly at L 93, "was" suggests that the area of the forest is not 1687 ha anymore now that the manuscript is submitted. Past tense is important in scientific papers, but only appropriate where events / methods / results really ARE past.

L100: (which are the even YEARS at our site) (clearer). Maybe it would be good to introduce this sentence with a more general one (I could only guess what you mean), e.g. something like "(This species of?) bamboo has alternating "off" and "on" years, in which the ... differs."

L113: Was this treatment repeated during the year, especially the growing season? If not, wasn't it necessary?

L117: How was the theta probe inserted? Permanently or for each of the measurements indicatd at Line 118? Horizontally or vertically?

L118: I do not understand. Typically the automated multiplexed system is used for continuous measurements, while "on selected days (about every two weeks)" sounds more like a manual survey measurement. How was the multiplexer exactly used - for all 4 plots or moving from plot to plot? What is meant by "mid-month"?

L133: then?

L134: grammar: reported as ... were chosen. Alspo try to avoid repeating "reported" between this and the next sentence, especiyll since they are about the same variable. Next sentence: For THE EC technique...

L138: How high was the bamboo canopy?

L140: Give more details on the processing - which corrections and quality checks? Which software?

L140 & 149: Not sure "whilst" is used appropriately.

L151: analysis was or analyses were

L160: "whilst" probably inappropriate

L162: mostLY

L164: shows

L166: exhibit instead of presented

Fig. 1: Replace "in long-term average" by "(long-term average)", drop sentence " legend shown as in figure"

Fig. 3: b also capital or all subfigure letters small

L186: Language example: "It (...)" - change construction.

L192: The word soil "surface" temperature seems to be newly introduced here. In the methods section the symbol Ts was used for a temperature measurement at 5 cm depth directly attached to the respiration chamber and I could not find TS5. I guess the Ts symbols with numbers refer to the Campbell array, but this does not seem to be explicitly mentioned and Ts and Ts5 is a bit strange to distinguish two temperature measurements that were both done at 5 cm.

L195-230: Do the R and R2 values refer to same regression analysis used to determine Q10 values (i.e. Rs was log-transformed) or are they strictly linear typical R values? Make clear.

L237: commonly what?

L236-263: The discussion of Ts-Rs relations at different depths, hystereses and phase shifts ignores the effect of heat transport in the soil, which has been abundantly discussed in the literature; one of the outcome of such studies was that the temperature measurement depth of highest R² does not necessarily indicate the prominent depth of respiration (as suggested at L249) or the "correct" Q10 value (e.g. Graf et al. 2008, Biogeosciences 5:1175).

L264-277: As far as I know the general expectation about SWC-Rs relations is that it is an optimum curve: At low to medium SWC, Rs depends positively on it because water is limiting, then there might be a plateau and at high SWC oxygen transport to the soil depth and transport of CO2 back might be blocked. Besides confounding effects with temperature which are briefly discussed but not throroughly checked, the reason for a nonexistent or negative linear correlation between SWC and Rs can therefore also be that the natural variation of SWC covers only a part of this curve.

L278: Unclear why the discussion now goes back to temperature relations, shouldn't this better be addressed directly with L236-263?

L281: No evidence is presented to support the hypothesis that soil moisture causes the different Q10 values in different seasons.

L329: Language example: "No obvious related..."?

L333-334: language example: "exhibited" twice close to each other.

Reviewer 3 ·

Basic reporting

a) English language should be deeply revised. Some parts of the manuscript are not easy to understand because of that. Just some examples in the abstract where the language could be improved: L21, L22, L24 or L27
b) References is one of the most important parts of a manuscript since establishes the foundation of the research. References should be carefully revised, since some of them are not related with the referred section (e.g. McDowell 2015 in L55)
c) The abstract should be improved and better organized. It directly starts with what the authors did, without introducing the state of the art or why was important measuring soil respiration in that forest. The goal of the study is not reflected in the abstract.
d) Introduction is well structured, but detailed goals of the manuscript are required. They sounds too general compared with results and discussion section.
e) Results and discussion section are confusing and topics order is not always logical. At some points, the reader could be lost.

Experimental design

Research presented in this study perfectly fits with the scope of the journal. Temporal variability of soil respiration and their relation with drivers has been largely studied for decades, and it does not seem that the way data is analyzed and presented is useful for filling a knowledge gap in the topic. Experimental design looks robust, but some important information on that section is missing (e.g. how different blocs are analyzed, how often Rs was measured (see specific comments below))

Validity of the findings

In my opinion, statistical approach was not the most appropriate for studying the aims of this study. I guess that the authors have good data for testing them, but the approach was not optimum. For instance, one of the aims of the study (distinguish the effect of different Rs drivers temporally autocorrelated) could not be tested just with direct correlations or multiple correlations. Interactions between drivers, such as SWC and temperature, should be tested. Additionally, relations between Rs and ecosystem productivity are poorly explored in the manuscript. I would suggest to the authors explore mixed effects models for analyzing the data. In addition, some of the conclusions are not well supported by the analysis.

Additional comments

In my opinion, the goals of the manuscript are interesting for the community and they have the enough data for addressing those goals. However, the used statistical approach and the way results are presented is not appropriate. The structure of the manuscript should be improved as well.

At certain points, results section looks like exploratory results. The authors showed a lot of figures (up to 24 panels), but many of them are not really related with the main goals of the manuscript (e.g. 2A, 2B, 2C, 3E). Accordingly, Discussion section is a little confusing and disconnected with the aims of the study.

Some specific comments for improving the manuscript:

L40 and L44-46. Why factors determining Rs are described in separated places? I guess they should be described together.
L49-51. Relation between productivity and Rs well demonstrated in many papers since decades, so this relationship is not a suggestion but a proved fact.
L55-58. This sentence is confusing.
L74-75. Redundant with the previous statement.
L87-89. I have never seen Effective accumulated temperature or Sunshine accumulated hours in soil respiration papers. Are these information relevant for the study?
L93. Flux tower? I guess you are talking about Eddy covariance flux tower. The way this is explained looks like the readers already know there was a tower, which is not the case.
L93-95. This sentence is confusing.
L106-107. LI-8150 is just a multiplexer. The authors should include information on the used analyzer (probably LI-8100?) and the chambers model.
L108. Which is the distance between plots? It seems that they are completely adjacent due to tubing limitations from the multiplexer. If so, why consider four plots with four replicates each one instead of 16 chambers randomly distributed nearby the tower?
L112. 24h after insertion is not enough for avoiding disturbance influence, considering that many fine roots may be cut in the first 5 cm of soil.
L114. The authors should explain this section in more detail. They did not say that they were performing automatic measurements.
L118-119. This should be better explained. It is still not clear when measurements were performed.
L132. When was LAI measured?
L144-145. Why the authors calculated Rs with temperature if they already measured Rs? This should be clarified.
L155. Results section is too long compared to other sections. Additionally, the way this section is explained is not fluent (e.g. As can be seen in figure 1…, Figure 2 shows…, Table 2 showed…, We plotted…). It might not be just a plain description of each figure and table.
L180-181. The statement exhibiting the typical growth characteristic of Moso bamboo in an “on year” needs a citation.
L183-186. How significance in diurnal patterns was tested. Figure 3A shows that Rs did not presented diurnal cycles.
L187-189. This statement describes seasonal variability rather than diurnal variations.
L190. Table 1 should be that one appearing earlier in the manuscript.
L193 and 221. The same thing explained at two different sites.
L194. See my comment on L144-145.
L211-219. This analysis should be explained in M&M section. Additionally, it is usually the interaction between SWC and temperature and not only their independently effect which controls RS.
L224. This should be explained in M&M section.
L246-251. This is not well supported by data.
L253-254. This could be just an artifact of measuring soil temperature at different depth from CO2 production.
L264-277. As mentioned before, the authors should test the interaction between SWC and temperature, and not the solely effect of each variable independently.
L278-279. This was not tested in the manuscript.
L316-322. This section is repetitive from the introduction.
Figure 5 is not explained. I don’t know what Rs residual means in Fig 5A or why they discard some points on Fig 5D.

---

## Round 0.2 · Minor Revisions

Thank you for your efforts in revising your manuscript and addressing reviewer comments. While the revised manuscript is improved, some additional comments have been provided by a reviewer to evaluated the original manuscript.

·

Basic reporting

I am happy with the response and the modifications in the revised manuscript.

Experimental design

It is now appropriate.

Validity of the findings

findings are valid for carbon cycle research in terrestrial ecosystems

Additional comments

See my response to the basic reporting section

Reviewer 3 ·

Basic reporting

I am glad to see how the manuscript has been improved since the last version and I hope my comments and suggestions could be used to improve the current version. Used English is much more better (this was not a limitation for understanding the manuscript) but I guess it still needs another revision to be published in a scientific journal (just talking about English). However, I am not a native speaker and perhaps I am not the best one for checking the writing. Literature and references seems appropriate.
In my opinion, there is too much information in the figures not relevant for the goals of the manuscript. There are 23 different panels and some of them are not relevant for the paper (even not described or discussed) and could be easily removed (see detailed comments at the end of the General Comments section). Additionally, a figure showing the seasonal course of soil respiration was not included, which might be relevant since is the main goal of the manuscript. The study includes all results relevant to the hypothesis.

Experimental design

The hypothesis is somewhat general (i.e. “explore the temporal dynamics of soil respiration, and to identify the relative importance of the measured environmental factors”); and because the hypothesis is too general, the results section and discussion are somehow vague and not always well oriented with different results not completely connected in the discussion.

Validity of the findings

I have two main comments for this section. First, the main conclusions of the study are that soil temperature and moisture are important drivers of Rs and that productivity and phenology could also play an important role. Despite there are a lot of studies reporting similar results for Rs, the authors mentioned that this had not been previously studied for Mozo bamboo forests. Secondly, most of the conclusions in the paper are based on causality inferred from direct correlations between pairs of variables. However, a lot of drivers tested in the manuscript showed strong autocorrelation (such as soil temperature, LAI, GEP), and thus causality cannot be directly extrapolated from correlations by pairs.

Additional comments

Soil and air temperature are strongly autocorrelated variables, and that is why Rs relationship with both of them are so similar. However, we already know that Rs is driven by soil temperature (and not air temperature), so including for all the analysis relationships between Rs and air temperature is redundant.
L37. Replace Bondlamberty for Bond-Lamberty
L51. Latitude is not actually a driver of Rs, but there are several variables influencing Rs that change with latitude.
L56-58. This is confusing. Which role they play in the context of sustainable development? Are they storing more carbon that woody forests, do they have higher growing rates and lower turnover? I would suggest removing this sentence since it is already explained further in the manuscript (e.g. L68).
L75. I would suggest scale back the tone of this sentence. An alternative would be “Thus, given the ecological importance of Moso bamboo forests at regional scale, there is a need for understanding the relationships between biotic and abiotic factors and Rs in this kind of ecosystem”.
L84. “… an eddy covariance flux tower…”
L88. “There are on average 2021 h of sunshine annually”. This information is somewhat strange. I suggest remove it.
L93. “The study area (1km around the eddy covariance flux tower) was covered by…”
L95. “the average canopy height was 11 m with a mean diameter at breast height of 9.3 cm”
L109. Replace “by the software of the manufacturer” by “(SoilFlux Pro, version…”
L120. I’m sorry, but I still found this a bit confusing. Were Rs only measured during daytime? In L108 you mentioned Rs was measured during 1 year, but here you said it was only during sunny days. Why did you remove the cloudy and rainy days? How do you think removing cloudy or rainy days would affect the calculation of annual Rs and your overall results, especially since you are studying the effect of moisture?
L125-131. Remove all information is not going to be used in the study. It is distracting. This also applies for the results section.
L127. “… close to the eddy covariance tower”
L146. Replace rejected by discarded
L150. See my comment on L 120.
L150-151. I do not understand point number 4.
L146-152. Are these QA/QC procedures commonly used for the eddy covariance community? If yes, provide a citation.
L154-155. Detailed information would be required here.
L156-158. This sentence could be removed since it is already explained previously.
L184-185. Do you mean warmest? It looks that several months were drier than July.
L193. Delete “value”
L202. It looks that for the most of seasons, there is no a diurnal cycle. How the authors tested for the significance of the diurnal cycles?
L209. I did comment on that at the last version of the manuscript. As far as I understood, you were using these equations for estimating the temperature sensitivity of Rs, not for estimating Rs. In fact, you don’t have to estimate Rs because you already measured it.
L210-211. Be careful with this kind of statements. Ts5 and Ta do not explain 69.7 and 63.9% of Rs variation. This are just correlations (or R2, to be more precise). Additionally, where did these numbers come from? They are mentioned even in the abstract and in the conclusions, but no figure or table seems to show them. Which statistical analysis has been made?
L216. I was wondering why the authors do not show a seasonal course of Rs. Instead of averaging the seasons as in Fig 3B you can show a kind of graph similar to those in Fig2A. At this point of the manuscript is still no clear how many days Rs was measured, so this figure could help on that.
L219-221. I would remove this sentence. It is a bit confusing and some of the information is already posted in the following sentence.
L222. Be aware that you are discarding all rainy and cloudy days, so your annual Rs might be overestimated.
L226. I have a couple of comments on this sentence.
a) In my opinion, it is still no clear the meaning of Fig5A. Which is the purpose of plotting LAI against residuals from Temp and SWC model? Do you want to test the effect of LAI on Rs without the effect of Temp and SWC? Doing that is challenging with variables that are strongly autocorrelated (such as Temp and LAI in your case), because you are overestimating the effect of temperature and thus, underestimating the effect of LAI. Additionally, which is the justification for using a parabolic fit in this figure? Which is the mechanistic explanation of a decrease in Rs when LAI goes higher than 4.5?
b) Temp, LAI and GEP are strongly auocorrelated (especially if GEP data from July and August are removed; Fig5D), so it’s logical that all of them showed a significant relation with Rs. Just be aware that direct causality cannot be inferred because of this autocorrelation.
L230-232. Which is the relevance of analyzing the relation between Rs and SWC if the effect of Temp*SWC interaction is already tested (Table 3)?
L238. Relation between Ts50 and Rs was not plotted.
L238. I guess that Fig3D is not relevant for the paper.
L239. Which is the difference between Ts and Ts5? (see L118 in the manuscript). Is Ts temperature measured close to each chamber and Ts5 temperature measured close to the tower? If yes, I think you should use the temperature measured close to the chambers regardless of the R2 they have with Rs.
L248-249. I am sorry, but I guess this statement is not fully supported by the results. As I mentioned before the autocorrelation between Temp, LAI and GEP is so high (as demonstrated in Table 2) that no direct conclusions can be taken from pairs of correlations.
L256-259. The lowest temperature sensitivity during summer was likely cause by low SWC values. During summer SWC became the limiting factor.
L261. I do not understand why hysteresis was mentioned here. Which is the relation between hysteresis and a decrease of temperature sensitivity during summer?
L280-283. I do not know if these potential explanations applies in your study case. Differences in photosynthetic supplies does not seem to be the reason because you have the larger hysteresis at winter, when LAI and GEP are the lowest. Similarly, SWC during winter was not a limiting factor, so the critical value explanation does not seem plausible.
L288-294. As you mentioned, this was probably an artifact. Rs has a positive relation with temperature and SWC During summer time, SWC was lower when temperature was higher, which has sense since soils get drier at higher temperatures. Thus, increments in temperature still produce an increase of Rs, despite SWC decreased. So the apparent negative effect of SWC was more related to the relation between SWC and Temp than a direct effect of SWC on Rs.
L294-298. I guess this is much more simple. SWC does not have effect because there were enough water in soil during all the year (just a little limitation in summer), so the limiting factor in your system was mainly temperature. Water shows usually significant effect at semiarid and arid ecosystems where there is enough temperature but water is limiting.
L333-334. In fig5 you are reporting month averages, which is not short-term relationships (comparing with results showing in Hogberg et al 2008). However, the authors have continuous Rs and GEP data, so this could be tested.
L336-337. Again, there is a strong correlation between LAI and temp which precludes the direct extrapolation of causality from correlations as you mention in the following sentence. Accordingly, “LAI reflected the productivity of vegetation” is not an accurate statement.
L349. The daily cycle of Rs has not been statistically tested and based on visual estimation from Fig3A is does not seem likely to be significant.
L351. In some parts of the manuscript, the authors said that temperature can explain 91.5% of Rs variability (Fig3C) and in other side it is said that temperature can explain 69.7%. Please, explain the difference and be consistent.
L352. Rs correlated negatively with SWC in summer (L231)
L353. Where did you test the diurnal effect of SWC on Rs?
L354-356. I do not understand this statement and I am lost finding the connection with the first part of the sentence. Additionally, I think this is too speculative to be in the conclusions of the manuscript.
Figures:
In my opinion, there is too much information in the figures not relevant for the goals of the manuscript
Fig2. SWC at different depths (B), NEE and RE (C) are not discussed in the paper. Consider remove them.
Fig3. Information placed in Fig3B is redundant with Fig3A, and Fig3D is not relevant for the study.
Fig4. As far as I understood, Ts5 was soil temperature measured at 5cm depth close to the EC tower and Ts was soil temperature measured close to each chamber during Rs measurement. I don’t know which extra information provides plotting hysteresis with the two soil temperatures. Since Ts was measured simultaneously with Rs close to the chamber, I would assume that Ts is more accurate and thus, I’d keep it. Accordingly, I would remove panels B, D, F and H.
Fig5. I still don’t understand panel A (see my comment above), and panel E has not been discussed in the entire paper. Additionally, revise the writing in the figure caption.
Table 3. It is the first time I’ve seen some of the equations you presented here for testing the relation between Rs and SCW and temperature [e.g. Rs=exp(a+b*Ts+c*SWC+d*Ts*SWC)]. Could you provide an explanation or a citation justifying why these equations have been selected?

---

## Round 0.3 · accepted · Accept

Thank you for your efforts in revising the manuscript and for your patience.

#